# Integrity Challenges in Halal Meat Supply Chain: Potential Industry 4.0 Technologies as Catalysts for Resolution

**DOI:** 10.3390/foods14071135

**Published:** 2025-03-25

**Authors:** Rizwan Matloob Ellahi, Lincoln C. Wood, Moin Khan, Alaa El-Din A. Bekhit

**Affiliations:** 1Department of Food Science, University of Otago, Dunedin 9016, New Zealand; rizwan.matloob@postgrad.otago.ac.nz; 2Management Department, University of Otago, Dunedin 9016, New Zealand; lincoln.wood@otago.ac.nz; 3The Federation of the Islamic Associations of New Zealand, 463 Porchester Road, Randwick Park, Auckland 2105, New Zealand; moin.khan@fianz.com

**Keywords:** halal, 3D printing, blockchain, Internet of Things, augmented reality, artificial intelligence, metaverse, digital twins, robotics, big data analytics

## Abstract

The application of Industry 4.0 technologies in the halal meat supply chain (HMSC) is an emerging but underexplored area. While technologies like 3D printing, AI, AR, metaverse, digital twins, robotics, and big data analytics are widely discussed in food production, their specific use in HMSC lacks comprehensive analysis. These technologies can address challenges such as cross-contamination, fraud, and traceability issues, but few studies have examined their practical implementation, highlighting the need for further empirical research. This review explores how Industry 4.0 technologies enhance efficiency, traceability, and transparency in HMSC and highlights the potential use of AR for real-time product verification, metaverse for virtual inspections, AI and robotics for improving efficiency, compliance, and hygiene, and digital twins for training and product quality monitoring. The review also identified research gaps, particularly the lack of focus on intelligent packaging, in vitro meat, and the 3D printing of halal meat products. The findings stress the need for greater exploration of these technologies in the HMSC, emphasizing their transformative potential in creating a transparent and efficient halal food system. Further research on emerging technologies like 3D printing and in vitro meat is essential, especially regarding their impact on halal standards and sustainability, ensuring future growth.

## 1. Introduction

Halal is an Arabic term frequently translated as “lawful or permissible” and is commonly associated with food, but the term in Islam goes beyond food and is essentially a way of life associated with gaining and managing the Lord’s blessings. The “halalness” of food, including meat, is a very important aspect for Muslims, encompassing criteria beyond the mere use of the ritual slaughtering method [1,2]. Recognizing the requirements for halal food production is crucial for the food industry to tap into this lucrative trillion-dollar market [3]. Muslims prioritize the halal status of meat in their purchasing decisions, often seeking certification and trusted approved logos for assurance. The production of halal meat entails stringent controls throughout the supply chain to ensure compliance, prevent contamination, and ensure wholesomeness. Nevertheless, there are instances where industry negligence or adulteration has created distrust [4], highlighting the need for robust systems to be in place to combat fraud and maintain consumer trust.

### 1.1. Halal Economic Importance

Halal food production adheres to Shari’a (Islamic laws), as prescribed in the Holy Quran and explained in authentic Hadith “narrated Prophet Muhammad (PBUH) sayings”. Halal slaughtering, known as Zabh, involves a rapid severing of the four blood vessels in the neck with a sharp knife, reciting Tasmiya (declaring God’s name with sincerity and conviction as a form of obtaining permission from the creator and declaring the process is not an act of aggression) [1]. The slaughtered animal must be healthy and alive at the start of the Zabh, and the process is carried out swiftly and as painlessly as possible. Several requirements are important to implement for halal slaughtering related to lawful ownership and animal welfare (full access to water and food, gentle and unaggressive handling, freedom of fear, no exhibition of a knife or subjecting the animal to an area covered with blood while it is conscious) [5]. A key condition of the permission given to man to slaughter and eat the flesh of animals is to mention the name of God during the slaughtering (Quran 6:121; 22:36). According to Al-Qaradawi [6], “acknowledging God during the slaughter is tantamount to the slaughterer by first obtaining consent from his and the animal’s Creator to take the life of another creature, and it is a declaration on the part of the slaughterer that his act is not an act of aggression against the universe nor of oppression of the creature about to be slaughtered, but simply an act necessitated by a need fulfilled in the name of God”. Conversely, under some conditions, consumption is prohibited, such as animals that were dead before exsanguination or were slaughtered without mentioning God, or in which other deities or names other than that of God were professed during the slaughter, which will corrupt the heart and result in a loss of piety and acceptance of prayer [6].

According to the Halal Food Global Market Report 2023 published by the Business Research Company [7], the market size was projected to increase from 1300.75 billion USD in 2022 to 1501.5 billion USD in 2023, with a compound annual growth rate (CAGR) of 15.4%. Several factors contributed to that increase such as the growth in the Muslim population and increased demand for halal-certified food due to the assurance of food safety and hygiene [8].

Despite its burgeoning value, the halal industry has not yet reached its full potential, with several areas offering opportunities for improvement, as indicated in the Global Islamic Economy Indicator (GIEI) by the DinarStandard report [9]. The halal food industry experienced significant advancements in 2018/19, driven by strong consumer spending and supported by trade initiatives and national economic strategies. Notable investments have also contributed to this progress. Despite these developments, the industry continues to face challenges in areas such as operations, regulation, and ecosystem growth. Nevertheless, rising consumer demand offers considerable opportunities for investors, governments, and businesses. Consequently, there is untapped potential for maximizing the halal industry by enhancing the quality of the halal supply chain (HSC) by prioritizing traceability and applying strict controls in the HSC [9].

The halal meat industry faces economic challenges such as supply chain bottlenecks, communication gaps, and variability in the acceptance of halal standards. These challenges can impact the internal operations of producers and influence consumer purchasing decisions [10].

The halal philosophy, rooted in Islamic economics, provides a framework for ethical and moral decision-making, which is integral to the economic activities of Muslims. This framework guides the behavior of the community and influences economic growth and community welfare [11,12].

While the importance of halal regulations is evident in ensuring product integrity and consumer trust, the economic insights reveal challenges that need to be addressed for the halal meat industry to thrive. The heterogeneity of halal standards and the lack of a unified regulatory framework pose significant hurdles for producers and consumers alike. However, with concerted efforts from regulatory bodies, industry stakeholders, and governments, these challenges can be mitigated to foster a more robust and competitive halal market.

### 1.2. Halal Meat Supply Chain (HMSC) Concerns

Concerns arise among consumers because major halal meat suppliers, such as Brazil, New Zealand, Australia, India, France, China, Netherlands, and Spain, are non-Muslim countries [13]. In 2020, Malaysia uncovered a meat-smuggling ring involving China, Ukraine, Brazil, and Argentina, exposing issues related to food safety and supply chain transparency [14]. This fake halal meat cartel scandal, revealed by a Malaysian newspaper, involved smuggling non-halal meat and highlighted significant concerns in the halal food industry.

The meat scandal involved mixing beef with diseased kangaroos and horses, which was falsely labelled as halal-certified. In Malaysia and Indonesia, harmful materials such as formalin, fake meat, exotic meat, dyes, and meat trimmings have been used to create fraudulent products. Adulterants are increasingly added to cut costs and increase quantities, and non-halal materials can be introduced due to poor handling and storage. The supply chains must be more transparent, traceable, and vigilant to prevent such issues [15]. Similarly, in Europe and the United States, cases of mislabeled non-halal meat as halal have raised concerns about consumer trust and the integrity of certification systems [16]. The issue arose when New York authorities found these products stored at non-refrigerated temperatures, contrary to their refrigeration requirements. Alef Sausage Inc. recalled 61,574 pounds of ready-to-eat halal meat and poultry sausage products due to misbranding and potential temperature abuse. For example, products like “Sheikh BEEF SALAMI ZABIHA HALAL” and others lacked proper handling statements, raising concerns about their safety if consumed without refrigeration [16].

As above-mentioned, halal meat fraud is a pervasive issue affecting various countries, which raises concerns about the integrity of certification systems and the need to implement recent advancements in the halal meat supply chain (HMSC) as crucial tools to maintain the authenticity of halal products worldwide.

### 1.3. What Is Meat?

The term “meat” commonly refers to animal muscle tissue that is consumed as food, and the term is loosely used to include organs and other animal parts that some cultural groups consume. The definition of meat may differ across contexts. For instance, the American Meat Science Association (AMSA) defines meat as “the skeletal muscle and its associated tissues from mammals, birds, reptiles, amphibians, and aquatic species commonly consumed by humans”. Boler and Woerner [17] highlighted that this definition is in contrast to those of other regulatory bodies like the USDA and ASTM International, which exclude certain animals such as fish, poultry, and hunted animals. Likewise, ASTM International characterizes meat as the muscle tissue of animals consumed as food, encompassing the processed flesh of cows, pigs, sheep, goats, and other consumable creatures, with the exclusion of fish, birds, and hunted animals. In contrast, AMSA does not recognize plant-derived, fungal, or microbial protein products as meat [17].

The historical practice of obtaining meat and other animal products through hunting has evolved into a modern system governed by stringent regulations that ensure humane slaughtering and hygienic processing practices, represented by the modern meat industry. Due to ethical and environmental issues that are thought to risk sustainable meat production, several new technologies to generate meat-like products (cultured meat, bioprinting, and plant-based meat) have received strong interest as future solutions to traditional meat production. A key question that is a matter of debate is whether these alternatives qualify to be defined as meat and if the definition needs to be revisited to accommodate novel meat alternatives. Furthermore, would the quality standards normally applied to meat and current traceability and risk management be suitable for these novel products? Currently, over 170 companies dedicated to cultivated meat as well as a rapidly growing number of scientists are actively engaged in the innovation and optimization of cultivated meat products. Their collective aim is to enable consumers to enjoy their preferred foods guilt-free, knowing that no harm to animals or negative impact on the environment was conducted in the production of the food. In 2023, a significant milestone was reached in the history of food and agriculture with the approval of the sale of a cultured meat product in the U.S., the largest world economy. This marked a turning point, as for the first time in history, a cultivated chicken product was served in U.S. restaurants, facilitated by two distinguished chefs [18]: Michelin-starred chef Dominique Crenn, who introduced Upside Foods’ chicken at Bar Crenn in San Francisco, while the acclaimed chef Jose Andres presented Eat Just’s Good Meat chicken at one of his establishments in Washington, DC. Furthermore, lab-grown meat has expanded into new venues in Singapore, featuring the first butcher shop where it was showcased and served in the adjacent café [18]. Investors are optimistic about the recent approval for U.S.-based companies Good Meat and Upside Foods to enter the market by the USDA and FDA. However, the adoption and growth of cultured meat face significant challenges, mainly due to the high costs. Over the past five years, billions have been invested in developing over 100 startups in the industry, with much more funding needed [19]. Despite regulatory progress, scaling production remains a significant and costly hurdle, as highlighted by the investor Lisa Feria, CEO of Stray Dog Capital [19].

In September 2023, JBS Biotech Innovation Center began building a 62 million USD R&D center for cultivated meat in Brazil. In 2022, Tyson Foods joined a 36.5 million USD Series A round for cultivated meat company Omeat, which launched in 2023. Tyson had previously invested in Upside Foods and Believer Meats. By January 2024, new global partnerships emerged to advance cultivated meat. Major CPG companies like JBS, Tyson, Cargill, Nestlé, and Danone are also involved in the industry through investments, acquisitions, partnerships, and R&D efforts [18].

The cellular production of meat is revolutionary compared with the conventional production of meat in several aspects, such as there is no slaughtering, rigor development (stiffness of the carcass after death], and conversion of muscle to meat process, which is very important to develop the essential characteristics of meat. Furthermore, the process lacks several issues that are required for halal meat production, such as the animal being alive, the Zabh (slaughter of the animal), Tasmyiah, and the complete drainage of blood, which creates doubts about the halal status of products produced from cellular meat. With the expansion in cellular food production and its perceived environmental and sustainability benefits [20], systems required for verification and traceability are needed if it enters the world meat trade chain, and more so, the ability to trace and differentiate it from conventional meat.

### 1.4. Advances in HMSC

Advancements in technology within the halal food sector are gaining momentum in response to the increasing demand for halal products. Several halal research centers have been playing a proactive role in devising methods for detecting components that diminish the halal status of food (e.g., porcine products or alcohol). Recent technological innovations such as blockchain (BC) and the Internet of Things (IoT) can provide new avenues to support the HSC. For instance, in August 2022, Fluree PBC, a U.S.-based open-source semantic graph database company, partnered with Sinisana Technologies, a Malaysia-based Web3 BC Traceability firm, to empower a Malaysia-based supply chain organization to provide unmatched end-to-end halal food product sourcing, provenance, and safe to its customers. Sinisana Technologies leverages technology in its product to ensure transparency and integrity throughout its supply chain and logistics operations, covering the entire life cycle of halal food items from their origins to grocery store shelves [8].

The first three industrial revolutions were marked by distinct technological leaps: steam power, electricity, and computers, where these advancements fueled mechanization, mass production, and automated manufacturing, respectively. Industry 4.0 (I4.0), introduced by Germany in 2011, ushered in a new era where physical and digital worlds merge. This convergence, known as cyber-physical systems (CPSs), allows for seamless communication between machines and virtual systems and creates a network of interconnected devices, enabling real-time access to information and services [21,22].

However, Industry 4.0 goes beyond automation and encompasses a vast array of innovations including BDA, cloud computing, smart systems, blockchain, artificial intelligence (AI), and the Internet of Things (IoT) [23,24]. Key Industry 4.0 Technologies [25,26,27,28] include:AI: Machines mimic human intelligence, enhancing decision-making and automation;Cloud computing: Offers remote computing power and scalable data management;3D printing: Creates physical objects from digital models;Augmented reality (AR): Overlays digital information onto the real world for immersive experiences;Robotics: Automates tasks in manufacturing and logistics;IoT: Connects smart devices for real-time communication;Big data analytics: Uncovers valuable insights from complex datasets;Blockchain: Provides secure and transparent transactions;Metaverse: Creates virtual environments for design, testing, and user interaction.

These technologies, along with others, such as drones and unmanned vehicles, hold immense potential to transform various industries [25,26,27,28].

The adoption of Industry 4.0 technologies in the halal industry remains in its early stages, with multiple barriers hindering widespread implementation. Despite the potential benefits of digital transformation, including enhanced efficiency and traceability, the halal industry has encountered significant challenges related to technological integration, stakeholder engagement, and awareness [29]. These challenges are further complicated by the necessity to align digital advancements with halal principles and the specific requirements of the halal supply chain.

One of the critical issues is the lack of comprehensive models that effectively integrate halal SMEs into the digital landscape. The absence of such models restricts SMEs from fully engaging with stakeholders and fulfilling operational requirements through digital solutions [29]. Additionally, the halal industry faces technological immaturity and limitations such as the underdevelopment of IoT devices and the high costs associated with implementing advanced technologies like blockchain and AI [30]. These limitations pose significant barriers to adopting Industry 4.0 technologies in the halal supply chain.

Moreover, effective coordination among halal business functions and stakeholders is essential for successful digital transformation. However, a lack of commitment and collaboration, especially among small- and medium-sized enterprises (SMEs), impedes the seamless integration of Industry 4.0 technologies into the halal ecosystem [29]. Addressing these challenges ensures that the halal industry can leverage digital advancements for improved efficiency, transparency, and sustainability.

## 2. Research Problem, Objectives, and Methodology

The application of Industry 4.0 technologies in the halal meat supply chain (HMSC) is an emerging but underexplored area. While technologies like 3D printing, AI, AR, metaverse, digital twins, robotics, and big data analytics are widely discussed in food production, their specific use in HMSC lacks comprehensive analysis. These technologies can address challenges such as cross-contamination, fraud, and traceability issues, but few studies have examined their practical implementation, highlighting the need for further empirical research.

A critical aspect of HMSC is maintaining integrity, which includes preventing fraud, ensuring compliance with the halal certification requirements, and safeguarding against cross-contamination. However, studies have rarely addressed how Industry 4.0 technologies can mitigate risks related to mislabeling, improper stunning and slaughtering practices, and the infiltration of non-halal elements into certified products. Addressing these integrity concerns is crucial for fostering consumer trust and regulatory confidence in the halal meat industry.

Despite the findings in the literature, discourse on Industry 4.0 remains restricted, particularly regarding its integration with the HSC and integrity considerations [31]. While scholars are actively engaged in various research domains related to Industry 4.0, endeavors to refine its definition and measurement often go unnoticed [32]. The lack of empirical evidence presents a significant challenge in developing comprehensive frameworks that connect Industry 4.0 with supply chain management, especially in the context of the HMSC and the critical integrity issues required to achieve diverse performance outcomes.

Thus, this review aimed to thoroughly investigate the challenges commonly faced within the HMSC. Additionally, it sought to evaluate the effectiveness and degree to which Industry 4.0 technologies can mitigate these challenges. The study also aimed to identify the role of emerging and underutilized technologies essential for facilitating a smooth transition toward Industry 4.0 in the halal meat industry supply chain.

### 2.1. Search Strategy

The database Scopus is recognized as the largest repository of peer-reviewed research publications and was chosen to retrieve pertinent published reports. The search criteria on Scopus were tailored to encompass the title–abstract–keywords.

The search criteria were devised to focus on emerging Industry 4.0 technologies reshaping supply chains. For a clearer understanding of the search parameters, Table 1 displays the search strings.

### 2.2. Inclusion Criteria

The table shows that as of 10 February 2024, 158 records were identified in Scopus using the search string. However, only records written in English that discussed halal food and applications of Industry 4.0 technologies were selected for the final inclusion. The definitive list of papers included in this study is presented in Table 1.

Notably as depicted in Figure 1, the search yielded limited results, with only 158 documents specifically related to the keyword “halal” in the context of Industry 4.0 technologies. These 158 articles included subjects beyond halal food, such as halal marketing, halal travel, and halal finance. After screening and removing duplicates, 23 relevant articles specifically focused on halal food were reviewed.

## 3. Results

Aligned with the research objectives, this section offers insights into the identified issues within the HMSC and how the upcoming Industry 4.0 technologies address these issues.

### 3.1. Integrity Issues in the HMSC

To provide a deeper understanding of the integrity issues within the HMSC, the frequency of these issues was gauged by thoroughly reviewing the research papers based on the search strategy. Figure 2 shows the issues in the HMSC identified by the review and the frequency of discussing these issues. The most common issues scholars reported included cross-contamination, fraud, and falsification. The identified HMSC issues are further depicted in detail in Table 2 and discussed below. While the frequency of discussion in the literature provides valuable insights into the prevalence of integrity issues within the halal food supply chain (HFSC), it is essential to recognize that less-discussed issues may still pose significant risks to the supply chain. By identifying these overlooked areas, researchers can focus on addressing the gaps in knowledge and developing solutions to enhance the integrity and transparency of the HFSC.

As shown in Figure 2, only one article addressed livestock management risks such as compliance with halal feeding protocols and the requirement for medications and vaccines to be halal. While this issue may not have received extensive attention in the literature, its inclusion underscores its importance and suggests an area where additional research is needed to ensure compliance with halal standards. It is important to note that the absence of discussion by multiple researchers does not diminish the significance of an issue. Instead, it can highlight potential risk areas that warrant further attention and investigation. Similarly, lack of technology adoption emerged as a significant factor, with only a few researchers highlighting its importance.

Table 2 provides a detailed and systematic analysis of the recurring challenges within the halal supply chain. From cross-contamination to fraudulent practices, traceability gaps, and inconsistent standards, it is evident that addressing these issues requires collaborative efforts across stakeholders. Implementing advanced technologies, establishing global standards, and enforcing stricter regulations could significantly improve the integrity of halal food supply chains. Citations ensure the credibility of the findings, offering valuable insights for policymakers, researchers, and industry practitioners aiming to strengthen halal practices globally.

Cross-ContaminationCross-contamination remains one of the most frequently cited concerns across the halal supply chain, as highlighted by Akbar et al. [41], Rahim [42], and Sidarto et al. [44]. It arises from shared facilities, equipment, and inadequate segregation during handling, storage, and transportation. For instance, Mardiyah et al. [43] emphasized the contamination risks due to multiple handling points, while Chandra et al. [46] noted the challenge of avoiding contamination in the interconnected global supply chain.The findings from Imanto and Yazid [45] further revealed doubts on the halal integrity of raw materials and processing facilities that handled both halal and non-halal foods, underscoring the lack of strict adherence to halal-specific protocols. Rejeb et al. [30] also noted the high likelihood of contamination in the food supply chain due to poor segregation and handling practices.Traceability and Transparency IssuesTraceability is repeatedly emphasized as essential to ensuring halal integrity. Akbar et al. [41] and Surjandari et al. [48] identified the inability to track product origins and the lack of transparency in food supply chains as major challenges. Similarly, Chandra et al. [46] and Inayatulloh [50] pointed to unreliable documentation, complex supply chains, and outdated information systems that make it difficult for stakeholders to verify halal compliance.Slow and unreliable traceability systems, as observed by Rahim [42], further exacerbate consumer uncertainty and industry inefficiencies. These findings underscore the urgent need for advanced digital solutions, such as blockchain, to enhance traceability, as suggested by various researchers.Fraud or FalsificationFalsification of halal certifications, tampering with labels, and the misrepresentation of products were recurring issues across studies. Maidin et al. [53] and Sidarto et al. [44] highlighted fake certifications and the improper labeling of non-halal products such as pork sausages being mislabeled as beef.Rejeb et al. [30] and Imanto and Yazid [45] added that counterfeit halal logos and mixing halal and non-halal substances were also significant concerns. These practices not only undermine consumer trust, but also reveal critical gaps in the monitoring and enforcement mechanisms.Multiple InterpretationsVariations in halal certification standards across regions complicate the global halal supply chain. Maidin et al. [53] and Chandra et al. [46] both emphasized that different certification bodies imposed varying guidelines, leading to confusion for manufacturers and consumers alike. Mardiyah et al. [43] also noted that these variations hindered consistent assessments of halal food compliance.Chandra et al. [46] observed the lack of harmonization across certifiers, which underscores the need for globally accepted standards. While complete standardization may not be feasible due to cultural and jurisprudential differences, harmonized processes could significantly improve the reliability and consumer confidence.Non-ComplianceSeveral authors, including Yusof et al. [51], Akbar et al. [41], and Maidin et al. [53], highlighted non-compliance with halal standards as a critical issue. These studies identified violations such as using unaccredited slaughtermen, unsanitary handling, and the failure to sanitize equipment between use.Muhammad et al. [52] also emphasized the risks of non-compliance in slaughtering processes, which directly impacts the Halal status of meat. These findings indicate a need for stricter oversight and Halal guidelines adherence at all stages of the supply chain.Lack of Regulation and EnforcementMaidin et al. [53] and Chandra et al. [46] highlighted the lack of adequate regulation, especially in non-Muslim countries. The absence of stringent enforcement allows for unchecked certification services, further complicating the global halal market.Sidarto et al. [44] also pointed to resource and infrastructure constraints as barriers to effective monitoring. These challenges highlight the importance of regulatory harmonization and capacity-building initiatives to strengthen the halal supply chain.Data Management and Technology AdoptionOutdated data systems and the insufficient integration of technology into the halal supply chain were highlighted as key issues by Tan et al. [49] and Surjandari et al. [48]. Delayed updates, unorganized filing systems, and the lack of real-time information hinder effective monitoring and compliance verification.Inayatulloh [50] added that traditional systems fail to provide validated data, underscoring the urgent need for advanced solutions such as IoT, blockchain, and digital twins to modernize the supply chain.Livestock Management RisksRejeb et al. [30] emphasized that compliance with halal feeding protocols and ensuring that medications and vaccines are halal-compliant are crucial aspects of livestock management. Any deviation could compromise the halal status of the resulting products, presenting another layer of complexity in the halal supply chain.

### 3.2. Application of Industry 4.0 Technologies in Enhancing HMSC

In addition to identifying challenges that impact the HMSC, this review examined the application of various Industry 4.0 technologies within the halal food sector. Table 3 presents an overview of the application of Industry 4.0 technologies by multiple researchers, and Table 4, Table 5, Table 6, Table 7, Table 8, Table 9, Table 10 and Table 11 present detailed applications of these technologies. These technologies represent cutting-edge advancements in manufacturing and offer innovative solutions to address the identified challenges and enhance the integrity and efficiency of halal food supply chains (HFSCs).

A detailed discussion of these Industry 4.0 technologies is provided below to explore the potential impact of these technologies in mitigating the identified challenges and driving the halal food industry toward greater integrity, transparency, and sustainability. By examining real-world applications and emerging trends, our goal was to provide valuable insights into the transformative potential of these technologies within the context of the halal food sector.

Several researchers have mentioned technologies in their research papers without providing descriptions of their applications within the context of the HMSC (Table 3). Therefore, Table 4, Table 5, Table 6, Table 7, Table 8, Table 9, Table 10 and Table 11 illustrate the application of these Industry 4.0 technologies in the context of the HMSC.

#### 3.2.1. Augmented Reality (AR)

The integrating of augmented reality (AR) into the HFSC offers notable advancements in safety, efficiency, and management. By using AR wearables, workers receive real-time processing and inspection instructions directly in their field of view, significantly reducing errors and contamination, which helps decrease food recalls and enhances the overall safety during maintenance. AR also improves warehouse and inventory management by assisting workers in handling large inventories with VR devices and AR-based applications, streamlining operations. Furthermore, AR enhances training by eliminating the need for on-site trainers, thereby reducing the training costs and allowing for on-site or factory-based training. It also aids logistics by using AR projections to display picking points in warehouses, thus optimizing the sorting efficiency and improving the overall logistics operations. These advancements collectively contribute to a safer, more efficient, and cost-effective HFSC (Table 4).

#### 3.2.2. Robotics

The integration of robotics into the HMSC significantly enhances efficiency and productivity. Automation and robotics enable the faster production of higher-quality products at reduced costs, addressing labor shortages and mitigating safety issues such as contamination from human error. In particular, collaborative robots (COBOTs) bring added benefits by working alongside human operators. They accelerate workflows in a cost-effective and adaptable manner, providing flexibility and agility in manufacturing processes. This reduces the operational costs and improves the overall safety and product quality in halal food production (Table 5).

#### 3.2.3. Digital Twins (DTs)

Digital twin technology substantially benefits the HMSC by enhancing product development, testing, and distribution. They enable halal manufacturers to optimize and forecast food products and plant operations, quickly identifying and addressing quality or productivity issues. By simulating virtual production lines before physical implementation, DTs improve plant efficiency and mitigate risks associated with investment, ensuring accurate production planning and safeguarding brand integrity. Integrating DTs with IoT technologies provides real-time performance insights, allowing the continuous monitoring of facility operations. This combination helps maintain equipment health, reduces maintenance costs, and optimizes the overall production efficiency, ultimately contributing to a more effective and cost-efficient HMSC (Table 6).

#### 3.2.4. Big Data Analytics (BDA)

BDA enhances the HMSC by improving compliance, transparency, risk management, certification processes, manufacturing, and logistics. Integrating IoT, cloud, and AI technologies with data analytics in Malaysia ensures traceability and quality assurance throughout the supply chain. In Indonesia, systems like LODHalal use comprehensive data to predict halal status and reduce fraud. BDA also optimizes the certification process by analyzing real-time data to classify ingredients and products, streamlining approvals and ensuring compliance. In manufacturing, it ensures halal compliance by processing data on raw materials and production processes. For logistics, it detects and prevents contamination, ensuring the efficient and Sharia-compliant handling of inventories (Table 7).

#### 3.2.5. Artificial Intelligence

AI significantly enhances the HMSC by optimizing compliance, monitoring, cost reduction, production efficiencies, waste management, operations, and decision-making. AI-based vision inspection systems monitor the slaughtering process, ensuring compliance with halal standards by detecting critical indicators like identifying and verifying the correct cutting of the esophagus, though challenges such as occlusion and noise can affect the accuracy. AI and robotics improve the production efficiency and product quality at reduced costs, oversee the food safety protocols, and minimize waste. Advanced robots handle tasks like picking and packaging, reducing the risk of damage. AI-powered robots now manage primary operations, including sorting and preparing raw materials, and secondary tasks like cooking and assembling. Technologies like TORMA food sorters use sensors and cameras for precise sorting based on various criteria. Additionally, AI aids in price forecasting, inventory management, and understanding consumer demand, with tools like Symphony RetailAI enhancing transparency and reducing wastage through better forecasting and inventory control (Table 8).

#### 3.2.6. Cloud Computing

Cloud computing is pivotal in enhancing the HMSC by enabling real-time data sharing, optimizing audits and compliance, supporting analysis, improving operations, and ensuring proper storage, handling, and transport. Cloud platforms facilitate efficient data collection, storage, and dissemination by integrating with technologies like blockchain and IoT. Tools such as Taiwan’s iMeasure app enable the real-time monitoring of perishable foods, making data accessible across the distribution network. Systems like QuikHalal streamline halal auditing processes by storing and managing data via IoT. Additionally, cloud-based platforms employ advanced technologies for quality and safety analysis, enhancing food safety inspections. BDA further ensures halal compliance throughout manufacturing and logistics by processing relevant data to prevent contamination and improve the overall efficiency (Table 9).

#### 3.2.7. The Internet of Things (IoT)

IoT significantly enhances the HMSC by improving compliance through real-time data sharing, traceability, and monitoring. IoT technology, including RFID and smart sensors, ensures the verification of product origins, precise record-keeping, and overall safety. For instance, in halal flour manufacturing, sensors monitor color, specks, moisture, protein, and ash content to optimize manufacturing efficiency. IoT also aids in monitoring animals and supervising workers to ensure compliance with halal standards. RFID technology helps halal-certified suppliers and logistics providers verify the status and safety of raw materials, while smart sensors and cameras continuously monitor halal and food safety points, tracking production stages, shipping times, and temperatures to proactively identify potential issues.

Moreover, IoT plays a crucial role in certifying the halal status of products, improving storage, handling, and transport, and assisting in auditing processes. It enhances the integrity of certifications by employing RFID and NFC technologies, which provide higher data security, operational visibility, and reliable product identification. IoT enables real-time environmental monitoring, improving the integrity of halal products during transportation and handling. Additionally, IoT technologies like GIS integrated with RFID improve logistics planning, and tools like “Halal tracer” use GPS and geofencing to prevent cross-contamination. In livestock management, IoT ensures transparency, capturing data on farm management and animal welfare, thereby building consumer trust. Finally, IoT helps reduce fraud by enabling consumers to verify the authenticity of halal logos, enhancing certification processes and consumer confidence in halal food products (Table 10).

#### 3.2.8. Blockchain Technology

Blockchain technology significantly enhances the HMSC by ensuring the transparency, authenticity, and reliability of halal products. By integrating distributed ledgers and smart contracts, blockchain fosters seamless information flow and efficient processes from production to consumer purchase. This technology aids in building consumer trust, promoting sustainability, and gaining international recognition for halal brands. For instance, Sreeya Sewu, a poultry company in Indonesia, implemented a blockchain-based traceability system for its poultry, ensuring transparency in the slaughtering process and digitalizing information flow. Blockchain meticulously tracks the participants’ addresses and their supply chain journey, providing transparent and improved process flows that enhance the halal supply chain’s sustainability and consumer trust.

Moreover, blockchain tackles critical issues in the HFSC such as trust concerns, fraud, and the contamination of non-Sharia-compliant ingredients. The use of blockchain, coupled with IoT devices like sensors and RFID, helps verify product provenance and ensure Sharia compliance through automated data collection and recording. This technology also addresses challenges posed by the diverse landscape of halal certification, facilitating the harmonization of certification processes and standards. By providing reliable and transparent certification, blockchain enhances consumer confidence and lays the groundwork for mutual recognition among certifiers, making the halal certification process more robust and cost-effective (Table 11).

## 4. Discussion

Table 3 provides an overview of the potential utilization of various technologies in the HMSC. It is important to highlight that several researchers have listed the names of Industry 4.0 technologies without providing descriptions of their applications within the context of the HMSC. Therefore, it has resulted in a lack of citations regarding the application of these technologies in Table 4, Table 5, Table 6, Table 7, Table 8, Table 9, Table 10 and Table 11, as most of the researchers have mentioned the technology but have failed to depict the application of these technologies in the context of the HMSC.

This indicates a gap in the existing literature and implies that these technologies are still at an early or developing stage, and thus, there may not be many documented instances of their successful implementation yet. This absence highlights the need for more research and case studies on how these technologies are applied in real-world scenarios related to halal meat traceability. There is a clear variation in the technologies that have been applied/proposed to play a role in the HMSC, with BC, IoT, Cloud, and RFID being the major technologies of interest.

It is evident that although certain technologies have been acknowledged by researchers, their utilization within the realm of halal food remains largely unexplored. For example, big data and AI are frequently cited in publications, but only three research papers have investigated their application specifically within the domain of halal food. Moreover, there is a noticeable lack of application of certain emerging Industry 4.0 technologies in the HMSC; interestingly, the application of DTs, augmented reality, and robotics in the discourse surrounding the HFSC was only discussed by Ahmad et al. [29]. Interestingly, two key Industry 4.0 technologies (additive manufacturing (3D Printing) and the metaverse) have been entirely overlooked by researchers in the literature concerning the HFSC.

Considering the current prevalence of blockchain, IoTs, RFIDs, and cloud computing in the literature, the discussion section will concentrate on the unexplored Industry 4.0 technologies within the context of the HMSC. This includes 3D printing, AI, augmented reality, metaverse, DTs, big data, and robotics.

As depicted in the review findings, cross-contamination, fraud or falsification, traceability and transparency, non-compliance, multiple interpretations, lack of regulation and enforcement, data management issues, lack of technology adaption, livestock management risks, and unreliable documentation are the most common issues in the HMSC. Industry 4.0 technologies are being applied globally to upgrade the resilience, reliability, traceability, and sustainability of supply chains globally [27].

### 4.1. Augmented Reality (AR) Technology in Addressing Problems in the HMSC

Our findings depict that augmented reality (AR) technology has emerged as a transformative solution in addressing critical challenges across the HFSC real-time processing and inspection instructions directly in front of workers. This capability reduces errors and contamination risks during production processes, ensuring strict adherence to halal standards, thus improving compliance and minimizing the potential for product recalls [29].

Pioneering a new era in dairy farm management, Nedap [77]. introduced the first augmented reality (AR) system specifically designed for this purpose. This groundbreaking technology leverages data from their award-winning Nedap CowControl system, a global leader in cow monitoring. Imagine a farmer’s world where crucial herd information comes alive right in the barn. Nedap’s AR system seamlessly overlays key cow insights onto the farmer’s real-time field of vision, providing the right information at the right moment. This intuitive system allows for natural interaction through hand gestures or voice commands, eliminating the need for cumbersome buttons or menus. By revolutionizing how farmers access and utilize data, Nedap’s AR solution empowers them to work more efficiently, productively, and ultimately, achieve greater success in their dairy operations. The industry recognized this innovative approach, receiving both a EuroTier Innovation Award and an Innovation of the Year award at EuroTier 2018 [77].

This system goes beyond traditional cow sensors by projecting relevant data directly into the farmer’s field of view through AR glasses. Nedap’s existing sensor technology, combined with Microsoft’s HoloLens (a business-oriented AR headset), creates a comprehensive picture of cow health and behavior. These sensors, typically placed around the neck and legs, track the activity levels, heat detection, rumination, and eating patterns. Additionally, barn-mounted units monitor their location, allowing farmers to find specific cows quickly.

Similarly, Zhao et al. [78] introduced a novel method using augmented reality (AR) to help dairy farmers in pasture-based settings identify and locate specific cows within large herds. Their approach leverages GPS collars on the cows, and a mobile device equipped with a camera and GPS. By combining these data with computer vision (CV) and machine learning, our mobile AR application offers two key functionalities:Search by unique ID: Farmers can easily locate a specific cow using its unique identifier.Real-time cow information: When a cow is selected on the screen, its associated data, including potential behavioral and health metrics, is displayed directly onto the live video feed through AR.This real-time information overlay empowers farmers to manage their animals with a focus on welfare, health, and timely interventions. The proof-of-concept application demonstrates the exciting potential of AR in advancing precision livestock farming, particularly for large-scale pasture-based dairy operational decisions. For instance, they can see how changes in feed rations impact cow behavior, enabling farmers to optimize animal well-being and farm productivity.

Similarly, we propose that AR can provide stakeholders with real-time information regarding the origin, handling, and certification of halal meat. Users can access comprehensive data about the meat’s journey from farm to fork using an AR-enabled device to scan a product. Consumers can scan product labels with AR to view interactive content that verifies halal certification, displays processing methods, and provides information on ethical practices followed.

AR can be utilized to train employees on halal standards and practices through immersive, interactive simulations, ensuring that they are well-versed in proper handling and processing methods. Additionally, AR can offer real-time, step-by-step instructions to workers in the supply chain, ensuring compliance with halal standards during slaughtering, processing, and packaging. The documentation process can be streamlined with AR by overlaying necessary forms and checklists during inspections and audits, ensuring that all steps are correctly followed and recorded. Furthermore, AR can aid in ensuring compliance with local and international halal certification standards by providing instant access to regulatory information and guidelines.

These applications of augmented reality in HMSCs will address non-compliance, uniformity, and interpretation within the standards, resulting in more transparent, reliable, sustainable, and resilient HMSCs.

### 4.2. Metaverse in Addressing Problems in the HMSC

The metaverse is “a simulated digital environment that uses augmented reality (AR), virtual reality (VR), and blockchain, along with concepts from social media, to create spaces for rich user interaction mimicking the real world” [79]. The burgeoning metaverse presents a compelling avenue for growth within the halal industry, as highlighted by Susanti et al. [80]. This immersive virtual environment offers the potential to streamline halal certification processes. By leveraging metaverse services, consumers could verify a product’s halal status remotely, eliminating the need for on-site inspections at production facilities.

This aligns with the broader trend of online process simplification within the halal sector. Local and international halal scanner applications, such as Smart Halal, Scan Halal, and My Halal Scanner, are experiencing increased global adoption. These user-friendly tools empower consumers with readily accessible halal product information. Entrepreneurs benefit from streamlined registration procedures for halal certification through online platforms offered by regulatory bodies like Indonesia’s Ministry of Religion (BPJPH).

Beyond its role in consumer confidence and safety, halal certification demonstrably enhances product marketability. Research suggests that the utilitarian benefits of halal-certified food maximize satisfaction for both businesses and consumers [80]. Furthermore, integrating metaverse technology into the halal food industry can foster unparalleled transparency regarding ingredient origin and halal compliance. This aligns seamlessly with the Islamic principle of hifz al-mal (protection of wealth) within the maqashid sharia (objectives of Islamic law) [80].

The integration of metaverse technology within the halal industry represents a significant leap forward in enhancing transparency, efficiency, and consumer confidence. The metaverse can uphold Islamic principles by streamlining halal certification processes and providing easy access to product information while boosting marketability and consumer satisfaction.

### 4.3. Robotics in Addressing Problems in the HMSC

The integration of robotics, including collaborative robots (COBOTs), can contribute to the HFSC in many ways. As Ahmad et al. [29] stated, deterring non-compliance and contamination issues can be a major application of robotics in HMSC. Robotics significantly enhance efficiency and productivity in halal food manufacturing by automating production processes. This capability allows for the rapid production of high-quality products at lower costs, effectively addressing labor shortages and reducing dependency. Robotics minimize food safety risks by reducing the potential for human error during food handling and processing. This automation leads to fewer instances of contamination, thus ensuring the integrity and safety of halal food products [29].

As per the Islamic Services of America, the rise of robotics is significantly impacting halal production, bringing a wave of benefits for certifiers, businesses, and consumers. By automating repetitive tasks, robots accelerate processing and packaging while maintaining strict hygiene standards. This not only increases the production output, but also minimizes the risk of cross-contamination between halal and non-halal items. Furthermore, robotic systems equipped with advanced vision technology elevate quality control to new heights and can meticulously inspect products for defects and verify halal certification labels, ensuring adherence to stringent requirements [81]. This translates into a higher quality and more trustworthy halal product for consumers. Additionally, automation with robots streamlines the certification process by handling data analysis, inspections, and report generation. This frees up valuable time and resources for certification bodies and allows businesses to obtain halal certification more efficiently. Finally, the reduced reliance on manual labor due to automation leads to cost savings in training and labor expenses. Companies can then strategically redeploy their workforce toward roles that require human expertise and supervision or are less suited for automation. Robotic technology is revolutionizing halal production, fostering efficiency, hygiene, quality, and cost-effectiveness across the entire halal supply chain [81].

### 4.4. DTs in Addressing Problems in the HMSC

The implementation of DTs in halal food manufacturing presents substantial solutions to various supply chain challenges. DTs forecast and optimize halal food products or plant operations, allowing for the swift identification and resolution of quality or productivity concerns. Integration with IoT provides real-time performance insights, enabling the continuous monitoring of facility operations and maintaining equipment health [29].

Even though the application of augmented reality, robotics, and DTs was only mentioned by Ahmad et al. [29] in our review, we suggest potential HMSC applications of DTs that can be further explored. This could lead to better monitoring, auditability, and accountability of HMSC operations. We suggest that further work can be undertaken to explore how a digital twin could be a virtual representation of each animal throughout the supply chain. This digital twin would be linked to real-time data on the animal’s origin, feed, medications, and slaughter methods.

Additionally, sensor data integrated with the digital twin could monitor animal welfare conditions like temperature, stress levels, and access to food and water. Cai et al. [82] discussed the use of virtual tools to simulate their physical counterparts safely and cost-effectively, though accurately replicating the physical tool behavior remains challenging. Engineers can utilize DTs to diagnose and predict issues when physical tools malfunction by integrating manufacturing and sensory data to enhance the accountability and capabilities of digital twin virtual machine tools in cyber-physical manufacturing. We believe a similar application with the application of DTs and their integration with sensory data can lead to more reliable and efficient HMSCs.

Roberts et al. [83] proposed a digital twin model to manage complex building structures and government pressures, providing a clear overview of compliance and accountability for existing buildings. The model aimed to demonstrate accountable outcomes, time and cost savings, and the achievement of goals like customer safety and asset strategy, with the potential for the scalability and IoT sensor integration for better insights. Similarly, Perez and Korth [84] proposed a digital twin model for ongoing auditing and evaluating legal responsibilities. The digital twin acts as a primary data repository, streamlining the creation of adherence guidelines. An event system connected to the digital twin classifies the status and alerts pertinent parties when necessary. Developing strategies to manage these findings is crucial for an effective legal compliance support system. Increasing demands from governments and customers for stricter monitoring are driving the need for digital support systems to ensure regulatory compliance.

As Chandra et al. [46] highlighted, different certification bodies follow distinct halal guidelines and charge varying fees, causing confusion among food manufacturers. Farrukh Habib, a researcher at ISRA and senior advisor at Halal Chain, recognizes the difficulties in achieving global standardization due to diverse jurisprudential, cultural, and behavioral differences. He opposes complete standardization and advocates for harmonizing various halal certification processes and standards [46].

Based on the above-mentioned applications of DTs, we suggest that the halal meat industry could benefit from similar applications of DTs embedded with sensors, which could address issues such as a variety of halal meat standards, multiple interpretations, compliance, and a lack of regulation and enforcement within the HMSC. The implementation of DTs in halal food manufacturing offers promising solutions for overcoming supply chain challenges by enabling the real-time monitoring and optimization of operations. This technology enhances auditability, accountability, and efficiency, potentially revolutionizing the HMSC through improved transparency and adherence to halal standards.

### 4.5. Big Data Analytics (BDA) in Addressing Problems in the HMSC

Big data technologies are instrumental in addressing key challenges within the HMSC. Multiple interpretations of halal standards are standardized through comprehensive data analysis, ensuring clarity and consistency. Real-time monitoring and analytics help enforce strict compliance with halal regulations, detecting and flagging any deviations promptly. Automated checks enhance regulatory enforcement, mitigating issues related to the lack of regulation and ensuring adherence to halal practices. Integration of IoT and data analytics enables the complete traceability of halal products, from sourcing to distribution, fostering transparency and accountability. Continuous monitoring and insights from big data maintain high standards of quality assurance throughout production. Advanced analytics and sensor technologies detect anomalies in handling and processing, safeguarding against non-compliance risks. Systems like LODHalal verify certification authenticity, reducing the prevalence of counterfeit halal products. Furthermore, big data accelerates the halal certification process by swiftly classifying ingredients and verifying Sharia compliance, enhancing efficiency and reliability in certifying halal products.

Rakhmawati et al. [85] proposed LODHalal, a linked open data (LOD) system specifically designed for halal food products. This innovative system addresses these limitations by utilizing standardized halal vocabulary and integrating data from diverse sources. LODHalal empowers consumers through user-friendly web and Android applications, offering valuable recommendations on the halal and nutritional content of food products.

Furthermore, LODHalal goes beyond basic verification. Developed in Indonesia, it provides a comprehensive information hub for consumers seeking clarity on a product’s halal status. The system gathers data from various trusted sources like DBpedia, MeSH, PubChem, and World Open Food Facts, with the crucial halal status information specifically obtained from LPPOM MUI, a respected Indonesian halal certification body. This comprehensive approach allows LODHalal to search for certified halal products and leverage BDA to predict the halal status of uncertified products with greater accuracy.

This innovative system offers a multitude of benefits. User-friendly web and Android platform interfaces make it convenient for consumers to access halal information on the go. Additionally, LODHalal empowers consumers by reducing the incidence of encountering fake halal logos and minimizing the risk of unknowingly consuming non-halal products. Ultimately, LODHalal fosters a more transparent and trustworthy halal food ecosystem for both consumers and businesses [85].

Big data technologies are transforming the HMSC by enhancing transparency, compliance, and efficiency through real-time monitoring and comprehensive data analysis. Systems like LODHalal leverage these technologies to ensure product authenticity and streamline the halal certification process, fostering a trustworthy and reliable halal food ecosystem for consumers and businesses.

### 4.6. AI in Addressing Problems in the HMSC

The integration of AI into the HMSC addresses numerous challenges, enhancing the overall efficiency, safety, and transparency. AI-driven vision inspection systems ensure proper halal slaughtering by accurately detecting critical indicators such as the esophagus in chickens, and overcoming issues like occlusion and varying illumination (Muhamad et al. [52]. AI also plays a crucial role in overseeing food safety protocols and minimizing waste during manufacturing. Advanced robots equipped with AI can adeptly handle and package irregularly shaped halal products, reducing damage and maintaining product quality. These robots can now perform primary manufacturing tasks like cleaning, sorting, inspecting, and transporting raw food materials as well as secondary operations such as blending, cooking, and assembling. AI facilitates the precise sorting of halal food products based on color, size, and shape, utilizing advanced sensor technology and adaptive spectrum cameras [29].

Furthermore, AI technologies, like those offered by Symphony RetailAI, assist in price forecasting, inventory management, and understanding consumer demand, thereby avoiding surplus and reducing wastage. By enhancing transparency and tracking in the halal certification process, AI builds greater trust among consumers and stakeholders, ensuring a more reliable and efficient HMSC [29].

As per the Islamic Services of America (ISA), halal consumers increasingly demand transparency and product authenticity. AI-powered solutions address this by leveraging advancements like visual inspection systems and the IoT to provide verifiable records throughout the supply chain, from ingredients to delivery. This not only validates the company’s halal business practices, but also safeguards against fraud. AI goes beyond verification, personalizing the customer experience through 24/7 AI-powered chatbots and tailored product offerings based on dietary and regional preferences. This fosters stronger customer relationships [86,87]. Maintaining halal compliance is simplified with AI continuously monitoring production lines and identifying deviations from standards. Visual inspection technologies ensure product authenticity, while AI empowers proactive measures to rectify potential issues [86,87]. AI fuels innovation by suggesting new halal product lines and analyzing consumer trends. Additionally, AI excels at gathering and analyzing market data, empowering data-driven decision-making for strategic marketing and brand building. By embracing AI, a Halal business can achieve continuous growth, adaptation, and success in the competitive marketplace, all while upholding the highest halal standards [86,87].

The integration of AI into the HMSC enhances efficiency, safety, and transparency, addressing challenges such as accurate halal slaughtering and food safety protocols. AI-driven technologies not only optimize manufacturing processes, like handling irregularly shaped products and precise sorting, but also support market insights and consumer demand forecasting, fostering a reliable and competitive halal industry.

## 5. Practical Implications

Integrating Industry 4.0 technologies holds profound practical implications for halal meat businesses, certification bodies, and policymakers. For halal meat businesses, advancements such as blockchain, IoT, and AI-driven automation can revolutionize supply chain transparency, simplify compliance procedures, and boost operational efficiency. Real-time monitoring of slaughtering practices, digital traceability of meat products, and automated quality control systems can minimize errors, mitigate fraud, and reduce risks of non-compliance. These improvements translate into cost savings, enhanced market competitiveness, and increased consumer trust in halal-certified products.

For certification bodies, Industry 4.0 technologies offer the opportunity to modernize verification and auditing processes, reducing the dependence on manual inspections. Blockchain can serve as an immutable digital ledger for halal certification, ensuring authenticity and curbing fraudulent practices. AI-powered image and video analytics can enable the remote monitoring of halal slaughtering practices, allowing certification bodies to oversee compliance more effectively, particularly in large-scale operations. Standardizing digital certification frameworks can also facilitate smoother international trade by harmonizing certification requirements across diverse markets.

For policymakers, adopting Industry 4.0 technologies in the HMSC can strengthen food safety, enhance regulatory compliance, and streamline international trade. Governments can develop digital certification platforms, promote the use of IoT-based monitoring systems, and implement AI-driven risk assessment tools to uphold halal integrity throughout the supply chain. Furthermore, policymakers can support businesses by offering incentives, funding research into halal-focused technological innovations, and fostering collaboration among technology providers, halal authorities, and industry stakeholders. By embracing Industry 4.0, policymakers can cultivate a more transparent, efficient, and globally recognized halal ecosystem, bolstering consumer confidence and expanding market opportunities for halal-certified products.

## 6. Limitations

While this review provides valuable insights into the potential applications of Industry 4.0 technologies in the HMSC, it had several limitations. It lacked a critical discussion on the feasibility and cost implications of implementing these technologies in real-world HMSC operations, particularly for SMEs. The financial burden of adopting technologies such as blockchain, AI, IoT, digital twins, and AR, along with the necessary training requirements for workers, remains unexplored.

While the study identified key challenges such as fraud and non-compliance, it did not sufficiently address the socio-cultural and ethical considerations of integrating AI, AR, and robotics into halal certification processes. These aspects are crucial for ensuring industry-wide acceptance and compliance with halal standards. The reliance on Scopus-indexed articles, omitting other relevant sources such as IEEE, Web of Science, and Google Scholar, may have introduced potential bias by overlooking critical studies from diverse research databases.

Furthermore, the paper did not provide a conceptual framework that integrated Industry 4.0 technologies into the HMSC. Although various technologies were mentioned, there was no structured model or set of recommendations outlining their implementation. A well-defined framework could offer a more strategic perspective on how these technologies could be effectively adopted across different supply chain stages.

Future research should focus on addressing these gaps by evaluating the financial and operational feasibility of Industry 4.0 technologies, exploring the ethical and socio-cultural dimensions, and developing a robust conceptual framework to guide their integration into the HMSC.

## 7. Future Directions

As outlined in the findings, numerous technologies have emerged as enablers for Industry 4.0. However, the literature on the HMSC lacks discussions on the application of such technologies. We advocate for the need to explore the utilization of emerging technologies such as 3D printing, metaverse, DTs, artificial intelligence, robotics, and augmented reality in modernizing the HMSC. Adapting to Industry 4.0 technological advancements is crucial as it represents the future, and the HMSC must align with these future-oriented directions.

Additionally, the growing concerns due to contamination and toxins surrounding packaging in the halal meat industry have sparked debate about the safety and integrity of the entire supply chain. Comprehensive measures are needed to ensure that packaging materials do not compromise the halal status of the meat. However, there is a potential gap, as most of researchers have neglected the packaging aspect and have failed to cater to intelligent packaging concepts for ensuring halal meat integrity.

Furthermore, there was a notable absence of discussion regarding the halal considerations of “in vitro meat”, also referred to as “cultured meat”, “lab-grown meat”, or “cell-based meat”. This innovative approach involves cultivating animal cells in a laboratory, diverging from traditional methods that entail the rearing and slaughtering of animals for meat consumption. Researchers have largely overlooked the multiple aspects of this invention, and there is a dearth of discourse regarding how the introduction of in vitro meat revolutionizes meat supply chains.

The existing literature lacks the application of 3D printing of meat and its implications on the HMSC. There is an emerging application of 3D printing methods in creating innovative food products, with a focus on integrating lab-grown meat or insect-derived ingredients. These applications aim to support ethical eating, improve food security, and contribute to environmental sustainability [88,89]. The advancing field of 3D printing (3DP) is recognized for its ability to produce intricately designed food items while minimizing the material costs and energy consumption. In the realm of sustainable 3D printing for meat alternatives, materials such as those derived from in vitro cell cultivation, meat by-products or waste, insects, and plants are being considered for their potential sustainability benefits. This innovation is reshaping traditional meat supply chains, necessitating significant research efforts to understand its implications, particularly in the context of HMSCs [88,89].

We perceive that the world is in a transitional phase, with less than half of the global population embracing Industry 4.0 technologies. Nevertheless, developed nations have made significant strides in embracing these advancements. It is crucial to synchronize with the technological progress of Industry 4.0, as it signifies the future trajectory.

## 8. Conclusions

This research thoroughly examined the traceability and transparency challenges prevailing within the HMSC industry. The findings underscored various integrity issues such as cross-contamination, fraud or falsification, lack of technology adoption, unreliable documentation or processes, traceability and transparency issues, data management issues, multiple interpretations, non-compliance, lack of regulation and enforcement, and insufficient infrastructure and resources, alongside livestock management risks.

The findings of this research will offer advantages to industries by shifting the focus toward neglected aspects such as technology adoption and halal-compliant livestock management. This, in turn, would enhance the broader infrastructure of HMSCs.

The findings have highlighted the emergence of numerous technologies as enablers for Industry 4.0, but there remains a significant gap in the literature regarding their application within the HMSC. We strongly advocate for researchers to delve into the utilization of emerging technologies such as 3D printing, metaverse, DTs, AI, robotics, and augmented reality to modernize the HMSC. Embracing these Industry 4.0 technological advancements is paramount as they represent the future, and the HMSC must align with these forward-looking directions.

Furthermore, the research highlights a significant oversight in the discourse regarding the halal considerations of “in vitro meat”, an innovative approach that involves cultivating animal cells in a laboratory setting. This revolutionary method has the potential to transform meat supply chains, however, there is a lack of discussion on its multiple aspects and how it reshapes the industry, but researchers have largely overlooked it. Halal meat regulations need to be created and expanded to include new meat alternatives such as lab-grown and plant-based meats, which challenge traditional definitions of halal. These regulatory frameworks must evolve to address the ethical, religious, and technological considerations that accompany such innovations.

The halal food industry is witnessing a revolution driven by a wave of innovative technologies. 3D printing, AR, metaverse, robotics, DTs, BDA, and AI are no longer futuristic concepts but powerful tools addressing critical challenges within the HMSC. This confluence of technologies fosters a more transparent, efficient, and trustworthy ecosystem for both consumers and businesses.

Consumers increasingly demand clarity and authenticity in their halal food choices. AR empowers them to take an active role in verifying product integrity. By scanning product labels with AR-enabled devices, consumers can access real-time data about the meat’s journey, from farm to fork. This includes information on its origin, handling practices, and most importantly, halal certification. The metaverse presents another exciting avenue for transparency. Virtual environments can streamline halal certification processes, allowing for remote verification and eliminating the need for on-site inspections at production facilities. This reduces costs and fosters trust by providing consumers with a virtual window into the halal journey of their food.

The integration of robotics brings much-needed efficiency and precision to the HMSC. Robots can automate repetitive tasks such as processing and packaging, minimizing human error and contamination risks. This enhances hygiene standards and reduces reliance on manual labor, addressing workforce shortages and streamlining operations. BDA empowers the real-time monitoring and optimization of production lines. By analyzing vast datasets, businesses can identify potential issues early on, ensuring strict compliance with halal regulations and preventing contamination.

Digital twin technology offers a groundbreaking solution for maintaining consistent quality throughout the supply chain. By creating a virtual representation of the entire ecosystem, from farm to processing plant, DTs enable the meticulous monitoring of animal welfare, product quality, and adherence to halal standards. Sensor data integrated with the digital twin can track factors like animal health, handling conditions, and processing temperatures. These real-time data allow for proactive interventions, ensuring the well-being of animals and the integrity of halal products. AI-powered vision systems further elevate the quality control. These intelligent systems can meticulously inspect products for defects and verify halal certification labels, ensuring adherence to stringent requirements.

AI does not just ensure quality; it personalizes the consumer experience. AI-powered chatbots provide 24/7 customer support, addressing inquiries and offering guidance throughout the purchasing journey. Additionally, AI analyzes the consumer preferences and dietary restrictions, enabling businesses to tailor product offerings and recommendations. This fosters stronger customer relationships and builds brand loyalty within the halal consumer base.

While the world is still in a transitional phase toward embracing Industry 4.0 technologies, developed nations have made substantial progress in this regard. It is paramount for the HMSC industry to synchronize with the technological progress of Industry 4.0, as it sets the trajectory for future innovation and advancement.

The integration of these emerging technologies signifies a paradigm shift in the halal food industry. By promoting transparency, efficiency, and quality, these advancements not only benefit consumers and businesses, but also contribute to a more sustainable halal ecosystem. Reduced waste, optimized resource utilization, and improved animal welfare are just some of this technological revolution’s positive environmental and ethical implications.

The adoption of Industry 4.0 technologies in the halal supply chain faces several barriers including high implementation costs, regulatory complexities, and resistance from halal certification bodies. The integration of advanced technologies such as blockchain, AI, and IoT requires significant financial investment in infrastructure, training, and system upgrades, which can be prohibitive for small and medium-sized enterprises (SMEs). Regulatory challenges further complicate adoption, as different countries have varying halal certification standards, making it difficult to implement a universally accepted digital solution. Additionally, halal certification bodies may be hesitant to embrace Industry 4.0 technologies due to concerns about automation replacing traditional verification methods, a lack of standardized digital frameworks, and the need for human oversight in religious compliance. Addressing these challenges requires a collaborative approach, including government incentives, industry-standardization efforts, and an increased awareness of how these technologies can enhance transparency, efficiency, and trust in the halal certification process.

In conclusion, the halal food industry is at the precipice of a transformative era. By embracing AR, metaverse, robotics, DTs, BDA, and AI, stakeholders can create a more transparent, efficient, and trustworthy HMSC that upholds the highest standards while catering to the evolving needs of consumers and businesses alike. This confluence of technology paves the way for a sustainable and prosperous future for the halal food industry.

## 9. Declaration of Generative AI and AI-Assisted Technologies in the Writing Process

During the preparation of this work, the authors used ChatGPT (Dunedin, New Zealand) to improve the readability and language. After using this tool/service, the author(s) reviewed and edited the content as needed, and take(s) full responsibility for the publication’s content.

## Figures and Tables

**Figure 1 foods-14-01135-f001:**
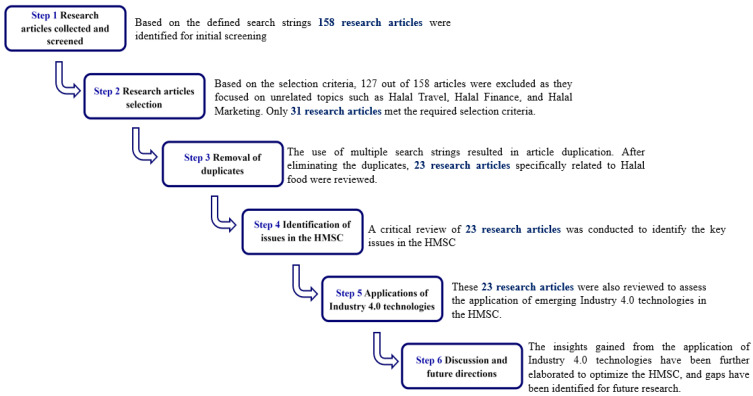
Research design of review “Integrity Challenges in Halal Meat Supply Chain: Potential Industry 4.0 Technologies as Catalysts for Resolution”.

**Figure 2 foods-14-01135-f002:**
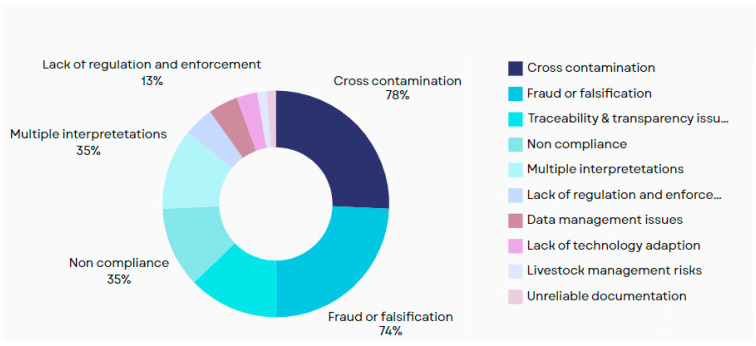
Frequency of the recurrence of these issues in the literature.

**Table 1 foods-14-01135-t001:** Number of articles retrieved based on each search string.

Access Date	Search String	Articles Found	Related to Halal Food
10 November 2023	(TITLE-ABS-KEY (artificial AND intelligence) OR TITLE-ABS-KEY (ai) AND TITLE-ABS-KEY (Halal))	30	3
10 November 2023	(TITLE-ABS-KEY (iot) AND TITLE-ABS-KEY (internet AND of AND things) AND TITLE-ABS-KEY (Halal))	15	2
10 November 2023	(TITLE-ABS-KEY (distributed AND ledger) OR TITLE-ABS-KEY (block AND chain) OR TITLE-ABS-KEY (blockchain) AND TITLE-ABS-KEY (Halal))	65	20
10 November 2023	(TITLE-ABS-KEY (big AND data) AND TITLE-ABS-KEY (Halal))	31	0
10 November 2023	(TITLE-ABS-KEY (digital AND twins) AND TITLE-ABS-KEY (Halal))	2	1
10 November 2023	(TITLE-ABS-KEY (augmented AND reality) AND TITLE-ABS-KEY (Halal))	8	3
10 November 2023	(TITLE-ABS-KEY (robotics) AND TITLE-ABS-KEY (Halal))	7	2
Total articles		158	31

**Table 2 foods-14-01135-t002:** Identified supply chain integrity issues in the HMSC.

Integrity Issue	Number of Authors Reporting	In-Text Citations
Cross-contamination	16	Wan-Chik et al. [33], Nizar, N. et al. [34], Hidayati et al. [35], Mulyaningsih et al. [36], Bin Khairuddin & Rahman [37], Susanty et al. [38], Ahmad et al. [29], Bin Illyas Tan & Hamid [39], Alamsyah et al. [40], Akbar et al. [41], Rahim [42], Mardiyah et al. [43], Sidarto et al. [44], Imanto & Yazid [45], Chandra et al. [46]
Fraud or falsification	14	Wan-Chik et al. [33], Nizar, et al. [34], Mulyaningsih et al. [36], Susanty et al. [38], Ahmad et al. [29], Bin Illyas Tan & Hamid [39], Rahman et al. [47], Alamsyah et al. [40], Surjandari et al. [48], Mardiyah et al. [43], Sidarto et al. [44], Imanto & Yazid [45], Rejeb et al. [30]
Lack of technology adaptation	4	Wan-Chik et al. [33], Nizar, N. et al. [34], Surjandari et al. [48]
Traceability and transparency issues	8	Nizar, N. et al. [34], Hidayati et al. [35], Bin Khairuddin & Rahman [37], Alamsyah et al. [40], Akbar et al. [41], Rahim [42], Chandra et al. [46]
Data management issues	3	Tan et al. [49], Alamsyah et al. [40], Inayatulloh [50]
Non-compliance	7	Susanty et al. [38], Ahmad et al. [29], Akbar et al. [41], Yusof et al. [51], Muhammad et al. [52], Sidarto et al. [44], Imanto & Yazid [45]
Multiple interpretations	7	Susanty et al. [38], Ahmad et al. [29], Maidin et al. [53], Surjandari et al. [48], Mardiyah et al. [43], Sidarto et al. [44], Chandra et al. [46]
Lack of regulation and enforcement	2	Maidin et al. [53], Chandra et al. [46]
Lack of infrastructure and resources	1	Sidarto et al. [44]
Livestock management risks	1	Rejeb et al. [30]

**Table 3 foods-14-01135-t003:** Identified Industry 4.0 technologies mentioned by HMSC researchers.

Technologies Mentioned by Researchers	BC (22)	IoT (14)	RFID (13)	NFC (3)	GPS (6)	QR (10)	Cloud (9)	AI (11)	BDA (5)	DTs (1)	AR (2)	Robotics (3)
Wan-Chik et al. [33]	x	x	x		x		x					
Nizar et al. [34]	x	x	x		x	x	x	x	x			
Hidayatia et al. [35]	x											
Mulyaningsih et al. [36]	x	x	x	x	x				x			
Tan et al. [49]									x			
Bin Khairuddin and Rahman [37]	x	x	x		x		x	x				x
Susanty et al. [38]	x	x	x			x	x					
Ahmad et al. [29]	x	x	x		x	x	x	x	x	x	x	x
Bin Illyas Tan and Hamid [39]	x	x	x			x	x	x	x		x	x
Rahman et al. [47]	x	x	x									
Alamsyah et al. [40]	x	x	x	x		x	x	x				
Akbar et al. [41]	x											
Rahim [42]	x						x	x				
Maidin et al. [53]	x											
Surjandari et al. [48]	x							x				
Mardiyah et al. [43]	x	x	x									
Sidarto et al. [44]	x					x						
Inayatulloh [50]	x											
Imanto and Yazid [45]	x	x	x			x						
Rejeb et al. [30]	x	x	x	x	x	x	x	x				
Yusof et al. [51]								x				
Chandra et al. [46]	x	x	x			x		x				
Muhammad et al. [52]								x				

**Table 4 foods-14-01135-t004:** Role of augmented reality in enhancing the HMSC.

Supply Chain Step	Impact of Augmented Reality	Description
Processing and Inspection	Error reduction and safety improvement	AR wearables display processing or inspection instructions directly in front of workers, reducing errors and contamination. Decreases halal food recalls, increases efficiency, and improves worker safety during maintenance [29].
Warehouse and Inventory	Efficient warehouse and inventory management	Assists workers in handling large inventories using VR devices and AR-based management applications [29].
Training	Improved training and safety	AR eliminates the need for trainers to be onsite, allowing for remote training. Reduces training costs and guides workers through maintenance training step-by-step [29].
Logistics	Optimized logistics	AR projections display the location of picking points in a warehouse to optimize picking activities. Halal manufacturers provide AR glasses to increase sorting efficiency [29].

**Table 5 foods-14-01135-t005:** Role of robotics in enhancing the HMSC.

Supply Chain Step	Impact of Robotics	Description
Processing and Production	Error reduction and safety improvement	Integrating automation and robotics enhances efficiency and productivity. Robots enable the production of higher-quality products quickly and at lower cost, addressing labor shortages. Robots mitigate safety issues for workers and reduce food safety risks caused by human error [29].
General Operations	Flexibility and cost efficiencies by collaborative robots (COBOTs)	COBOTs accelerate workflows in a cost-effective, safe, and adaptable manner. COBOTs introduce flexibility and agility, reducing costs for halal food manufacturers [29].

**Table 6 foods-14-01135-t006:** Role of DTs in enhancing the HMSC.

Supply Chain Step	Impact of DTs	Description
Product Development	Facilitates halal product development	DTs optimize halal food products or plant operations. Swift identification and resolution of quality or productivity concerns [29].
Planning and Risk Management	Better planning and risk management	Virtual production lines enhance plant efficiency and minimize risks before physical implementation. Predict future production or product scenarios without jeopardizing brand integrity [29].
Facility Operations	Improved safety and efficiency	Digital twin with IoT delivers real-time performance insights. Continuous monitoring of facility operations maintains equipment health and enhances efficiency. Lowers maintenance expenses and optimizes production efficiency [29].

**Table 7 foods-14-01135-t007:** Role of big data in enhancing the HMSC.

Supply Chain Step	Impact of Big Data Analytics	Description
Compliance and Transparency	Improves compliance and transparency	IoT, cloud, and AI integration with data analytics achieve traceability and transparency. Sensors and technologies monitor goods movement and ensure compliance with halal standards. Anomalies can be detected easily, and potential non-compliance is flagged [34].
Risk Management	Better risk management and fraud detection	LODHalal system in Indonesia provides comprehensive halal information and fraud detection, gathering data from various sources to predict the halal status of uncertified products [39].
Certification	Optimized certification process	Real-time data from manufacturers can classify ingredients as halal, haram, or doubtful, speeding up the certification process. Companies audited are categorized as pass, fail, or in progress, based on real-time data analysis of products, ingredients, production tools, and halal standards. BDA streamlines the certification process, enables faster approvals, and automatic updates to standards [49].
Manufacturing	Optimized manufacturing	BDA processes data on raw materials, packaging, suppliers, and ingredients for halal compliance, providing valuable information to expedite the halal certification process [49].
Logistics	Optimized logistics	BDA detects and prevents halal product contamination during delivery. Tracks daily logistics activities and identifies potential contamination [49].

**Table 8 foods-14-01135-t008:** Role of AI in enhancing the HMSC.

Supply Chain Step	Impact of AI	Description
Compliance and Monitoring	Optimized compliance and monitoring	AI-driven vision inspection systems enhance the monitoring of the slaughtering process by detecting key indicators like identifying and verifying the correct cutting of the esophagus, which is crucial for ensuring proper halal slaughtering. Techniques include image processing and detection to improve accuracy and address challenges such as occlusion and varying illumination [52].
Production Efficiency and Waste Management	Cost cutting, production efficiencies, and waste management	AI and robotics enhance halal production processes by improving efficiency and maintaining product quality while minimizing costs. AI oversees the food safety protocols and reduces waste during manufacturing. Robots handle tasks such as picking and packaging irregularly shaped products, reducing the risk of damage [29].
Operations	Optimized operations	AI and robotics expand roles beyond traditional tasks, including cleaning, sorting, inspecting, measuring, preparing, and transporting raw materials. AI facilitates sorting based on criteria such as color, size, and shape using sensor technology and adaptive spectrum cameras [29].
Analysis and Decision Making	Optimized analysis and decision making	AI-generated data support various functions such as price forecasting, inventory management, and understanding consumer demand. For example, Symphony RetailAI helps forecast pricing dynamics, transportation needs, and inventory levels, thus optimizing operations and reducing wastage [29].

**Table 9 foods-14-01135-t009:** Role of impact of cloud computing in enhancing the HMSC.

Supply Chain Step	Impact of Cloud Computing	Description
Data Sharing and Transfer	Enables real-time data sharing and transfer	Cloud computing integrates with various technologies like blockchain and IoT for data collection, storage, and sharing. The iMeasure app monitors perishable food temperature through wireless sensor tags across the supply chain [33]. Cloud platform integration with IoTs facilitates real-time halal information dissemination within internal operations or transportation chains, reducing the risk of contamination [37].
Audit and Compliance	Optimized audit and compliance	QuikHalal, a cloud-based mobile halal auditing tool, collaborates with IoT to store data, streamlining the planning, auditing, and reporting of halal audit compliance [37].
Analysis	Supports analysis	Manufacturers can leverage sensor technologies like near-infrared spectrometers and hyperspectral imaging to capture image data for subsequent cloud-based analysis using machine learning algorithms, enabling insights into product quality and safety [29].
Operations	Optimized operations	Halal products must be safe, traceable, and free of haram elements including ingredients, equipment, packaging, and additives. Cloud-enabled big data analytics (BDA) processes data on these elements to ensure halal compliance, product improvement, and faster certification [49].
Storage, Handling, and Transport	Improves halal product storage, handling, and transport	Halal inventory must adhere to Sharia law during procurement, movement, storage, and handling. Cloud-enabled BDA analyzes transportation data to detect and prevent contamination, optimizing logistics, and ensuring Halal product integrity [49].

**Table 10 foods-14-01135-t010:** Role of IoT in enhancing the HMSC.

Supply Chain Step	Impact of IoTs	Description
Compliance and Monitoring	Enhances compliance through real-time data sharing, traceability, and monitoring	IoT improves halal supply chain management through advanced tracing and tracking, ensuring product origin verification, precise record-keeping, and safety. Sensors in halal flour manufacturing monitor color, specks, moisture, protein, and ash content in real-time [54,55,56]). RFID technology verifies the halal status and safety of raw materials and ingredients [30,54,57]. The continuous monitoring of production stages, shipping times, and temperatures through smart sensors and cameras allows for proactive identification of potential issues and outbreaks, ensuring compliance with halal standards [30,54,57].
Certification	Certifies halal status of products	RFID technologies help certify the Halal status of food raw materials and ingredients [30,54,57,58,59]. RFID surpasses traditional tracking tools like barcodes for Halal food [60]. Enhances product operational visibility and provides dependable product identification [61].
Storage, Handling, and Transport	Improves halal product storage, handling, and transport	GIS and RFID integration improve planning and operations in halal food transportation, optimizing logistics, and loading/unloading processes [58]. Halal Tracer uses GPS and geofencing to monitor shipments, vehicle stops, and driver actions, preventing cross-contamination [62]. Developed by Taiwan’s Institute of Industrial Technology Research, the iMeasure app monitors the temperature of perishable food using wireless sensor tags and includes a cloud system platform for data visibility [63].
Auditing	Assists in auditing	QuikHalal, the world’s first mobile app for halal inspection developed in Malaysia, serves as a cloud-based mobile halal auditing application in collaboration with IoT for data storage. It effectively addresses documentation challenges during halal audits, providing a streamlined solution [39,64,65].
Livestock Management and Welfare	Improves livestock management and welfare	Records on-farm management, halal feeding protocols, animal welfare, health events, slaughter procedures, and product transportation to enhance transparency and consumer trust [55,66]. Utilizes 2G-RFID sensor tags for automatic livestock identification and health monitoring, ensuring data integrity and mitigating contamination risks [67,68].
Supply Chain Transparency	Enables end-to-end transparency in supply chain	RFID technologies streamline authentication processes for halal food. The combination of IoT technologies, RFID tags, and QR codes verifies the Halal status of products [30,69].
Fraud Reduction	Reduces fraud	Halal logo detector and RFID-based systems combat counterfeiting and bolster consumer confidence [70,71].
Certification Integrity	Enhances integrity of certifications	NFC technology authenticates halal certificates, streamlining compliance and monitoring [72]. Halal-Square, in collaboration with JAKIM, has developed the world’s first NFC-based halal certification system, which integrates with the Halal-Inside.com app. This system allows users to verify halal certificates via NFC-enabled phones, providing updates on halal status and certificate expiry dates [36,73].

**Table 11 foods-14-01135-t011:** Role of blockchain in enhancing the HMSC.

Supply Chain Step	Impact of Blockchain	Description
Information Transparency	Transparent information and improved process flow	Blockchain ensures halal supply chain transparency, fostering trust and efficiency through distributed ledgers and smart contracts. This technology enhances product traceability, supports sustainability, and strengthens halal brand recognition [74]. Sreeya Sewu, an Indonesian poultry giant, implemented a pioneering blockchain-based traceability system for its halal chicken slaughtering process, ensuring transparency and digitizing information flow [44].
Trust Enhancement	Enhances trust and confidence	Blockchain technology enhances the halal supply chain by transparently tracking transactions and supply chain journeys, fostering sustainability, consumer trust, and international recognition for halal initiatives [74].
Fraud Reduction	Reduces fraud and other concerns	Blockchain provides the halal industry with unparalleled opportunities to solve problems such as trust concerns, pollution control, fraud, falsification, and fake halal emblems in the supply chain and logistics by adopting this groundbreaking and disruptive innovation [43].
Compliance and Auditability	Improves compliance, auditability, and accountability	A blockchain framework enhances transparency, upholds halal integrity, and streamlines the halal certification process in a cost-effective manner [75]. A blockchain-based traceability system in halal labeling ensures transparency and accountability by using interconnected blocks containing data, a unique ID (hash), and a previous hash, making tampering difficult [76]. Blockchain, combined with IoT devices (sensors, RFID), records data and verifies product provenance and Sharia compliance, addressing challenges with poorly regulated raw materials [46].
Standardization	Supports uniformity	The global halal certification sector faces challenges in standardization with over 300 certifiers and varied standards. Dr. Farrukh Habib advocates for harmonizing certification processes and using blockchain technology to enhance transparency and mutual recognition [46].
Animal Welfare	Improves animal welfare	Blockchain technology ensures the transparency and traceability of animal welfare standards throughout the supply chain. It records data on animal treatment, health status, and welfare compliance, providing consumers with trustworthy information about the ethical treatment of animals in the halal supply chain [55,66]).

## Data Availability

Data sharing is not applicable to this article as no new data were created or analyzed in this study.

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
