# Peer review of "Integrity Challenges in Halal Meat Supply Chain: Potential Industry 4.0 Technologies as Catalysts for Resolution"

_foods, 2025, doi:10.3390/foods14071135_

Round 1

Reviewer 1 Report

Comments and Suggestions for Authors
  • The paper identifies and discusses the challenges in the Halal Meat Supply Chain (HMSC) and the potential use of Industry 4.0 technologies (Blockchain, IoT, Cloud Computing,  RFID technologies, Artificial Intelligence, Robotics, and Augmented Reality) to resolve these challenges and modernize the HMSC.
  • The Introduction provides relevant background and highlights some gaps in research, such as the lack of discussion on intelligent packaging's role in maintaining Halal meat integrity. It should state more gaps and discuss the significance of the study as well as the problem statement (it is discussed synoptically in Section 2 but it also needs to be discussed in the Introduction).
  • Fraud in the Halal Meat Supply Chain is repeated many times and in multiple sections. I suggest summarizing them in one place (Literature review and Conclusions and referring to them when needed).
  • The Conclusion section needs to discuss barriers to the adoption of these technologies (e.g., cost, regulatory issues, acceptance in Halal certification bodies).
  • Moreover, since fraud and traceability are the most critical success factors of the HMSC I suggest re-organizing the Conclusions part into two sub-sections and summarizing the effect of the examined technologies into the two concepts.
  • Also, a section on practical recommendations for Halal meat businesses, certification bodies, and policymakers would add value.
Comments on the Quality of English Language

The English could be improved to more clearly express the research.

Author Response

The authors would like to thank the reviewers for their time and comments that improved the work substantially.

The MS has been revised for language by a native-English-speaking co-author.

 Reviewer 1

Comment: The Introduction provides relevant background and highlights some gaps in research, such as the lack of discussion on intelligent packaging's role in maintaining Halal meat integrity. It should state more gaps and discuss the significance of the study as well as the problem statement (it is discussed synoptically in Section 2 but it also needs to be discussed in the Introduction).

Thank you for your comment. We have carefully considered your comment and have added a following in lines # 236-255 to address the comments “The adoption of Industry 4.0 technologies in the halal industry remains in its early stages, with multiple barriers hindering widespread implementation. Despite the potential benefits of digital transformation, including enhanced efficiency and traceability, the halal industry encounters significant challenges related to technological integration, stakeholder engagement, and awareness (Ahmad et al., 2022). These challenges are further complicated by the necessity to align digital advancements with halal principles and the specific requirements of the halal supply chain.

One of the critical issues is the lack of comprehensive models that effectively integrate halal SMEs into the digital landscape. The absence of such models restricts SMEs from fully engaging with stakeholders and fulfilling operational requirements through digital solutions (Ahmad et al., 2022). Additionally, the halal industry faces technological immaturity and limitations, such as the underdevelopment of IoT devices and the high costs associated with implementing advanced technologies like blockchain and AI (Rejeb et al., 2021). These limitations pose significant barriers to the adoption of Industry 4.0 technologies in the halal supply chain.

Moreover, effective coordination among halal business functions and stakeholders is essential for successful digital transformation. However, a lack of commitment and collaboration, especially among SMEs, impedes the seamless integration of Industry 4.0 technologies into the halal ecosystem (Ahmad et al., 2022). Addressing these challenges is crucial to ensuring the halal industry can leverage digital advancements for improved efficiency, transparency, and sustainability.”

Comment: Fraud in the Halal Meat Supply Chain is repeated many times and in multiple sections. I suggest summarizing them in one place (Literature review and Conclusions and referring to them when needed).

Based on your valuable comment, we have removed redundancy and have added rephrased version in line# 111-115 as “In 2020, Malaysia uncovered a meat-smuggling ring involving China, Ukraine, Brazil, and Argentina, exposed issues related to food safety and supply chain transparency (Ariffin et al., 2021). This fake Halal meat cartel scandal, revealed by a Malaysian newspaper, involved smuggling non-Halal meat and highlighted significant concerns in the Halal food industry.”

Comment: The Conclusion section needs to discuss barriers to the adoption of these technologies (e.g., cost, regulatory issues, acceptance in Halal certification bodies).

Based on your valuable comment, we have expanded the conclusion to reflect barriers to the adoption of these technologies

Comment: Moreover, since fraud and traceability are the most critical success factors of the HMSC I suggest re-organizing the Conclusions part into two sub-sections and summarizing the effect of the examined technologies into the two concepts. (discuss)

Done.

Also, a section on practical recommendations for Halal meat businesses, certification bodies, and policymakers would add value.

Based on your valuable comment, we have added section 5. as Practical implications

Thank you once again for your valuable feedback, which has helped us improve the quality of our work.

Reviewer 2 Report

Comments and Suggestions for Authors

The following comments are required to improve this paper.

Abstract

How does each technology contribute uniquely to HMSC?

Introduction

This part should be more strengthening coherence between economic insights and Halal regulations that would enhance readability and impact.

Background

The background provides strong evidence of supply chain issues but contains redundancy, particularly in repeating the 2020 Halal meat scandal. Refining the structure and eliminating repetition would enhance clarity and impact of present study.

Research problem, objectives & Methodology

This part effectively highlights gaps in integrating Industry 4.0 with HMSC but lacks specificity in defining integrity concerns.

Moreover, it is suggested that refining the research objectives and methodology links to empirical validation that would strengthen the study's contribution.

Further Remarks

The study provides a valuable exploration of Industry 4.0 technologies in the Halal Meat Supply Chain (HMSC), highlighting key gaps and potential benefits. However, it lacks critical discussion on the feasibility and cost implications of implementing these technologies in real-world HMSC operations, particularly for small and medium enterprises. Additionally, while the study identifies challenges like fraud and noncompliance, it does not sufficiently address the socio-cultural and ethical considerations of integrating AI, AR, and robotics into Halal certification processes.

The study primarily focuses on Scopus-indexed articles, omitting other relevant sources such as IEEE, Web of Science, and Google Scholar, this might lead to potential bias.

The paper does not provide a conceptual framework integrating Industry 4.0 technologies into the Halal Meat Supply Chain (HMSC). It only mentioned the technologies. At least there should be a framework or suggestions.

What are the cost implications of using the Blockchain, AI, IoT, Digital Twins, AR?

 What are the training requirements for workers? It should at least mention some guidelines and requirements

Comments on the Quality of English Language

The language may be improved further.

Author Response

The authors would like to thank the reviewers for their time and comments that improved the work substantially.

The MS has been revised for language by a native-English-speaking co-author.

 Reviewer 2

Comment: Indroduction:This part should be more strengthening coherence between economic insights and Halal regulations that would enhance readability and impact

Thank you for your comment. We have carefully considered your comment and have added the following in lines #86-99: “The halal meat industry faces economic challenges such as supply chain bottlenecks, communication gaps, and variability in acceptance of halal standards. These challenges can impact the internal operations of producers and influence consumer purchasing decisions (Jailani, 2024).

The halal philosophy, rooted in Islamic economics, provides a framework for ethical and moral decision-making, which is integral to the economic activities of Muslims. This framework guides the behaviour of the community and influences economic growth and community welfare (Jah & Tie, 2024) (Faozi et al., 2023).

While the importance of halal regulations is evident in ensuring product integrity and consumer trust, the economic insights reveal challenges that need to be addressed for the halal meat industry to thrive. The heterogeneity of halal standards and the lack of a unified regulatory framework pose significant hurdles for producers and consumers alike. However, with concerted efforts from regulatory bodies, industry stakeholders, and governments, these challenges can be mitigated to foster a more robust and competitive halal market.”

Comment: The background provides strong evidence of supply chain issues but contains redundancy, particularly in repeating the 2020 Halal meat scandal. Refining the structure and eliminating repetition would enhance clarity and impact of present study 

Based on your valuable comment, we have removed redundancy and have added rephrased version in line# 289  as “In 2020, Malaysia uncovered a meat-smuggling ring involving China, Ukraine, Brazil, and Argentina, exposing issues related to food safety and supply chain transparency (Ariffin et al., 2021). This fake Halal meat cartel scandal, revealed by a Malaysian newspaper, involved smuggling non-Halal meat and highlighted significant concerns in the Halal food industry.”

Comment: Research problem, objectives & Methodology: This part effectively highlights gaps in integrating Industry 4.0 with HMSC but lacks specificity in defining integrity concerns.

Moreover, it is suggested that refining the research objectives and methodology links to empirical validation that would strengthen the study's contribution.

Based on your valuable comment, we have revised the Research problem, objectives & Methodology sections.

Comment: The study provides a valuable exploration of Industry 4.0 technologies in the Halal Meat Supply Chain (HMSC), highlighting key gaps and potential benefits. However, it lacks critical discussion on the feasibility and cost implications of implementing these technologies in real-world HMSC operations, particularly for small and medium enterprises. Additionally, while the study identifies challenges like fraud and noncompliance, it does not sufficiently address the socio-cultural and ethical considerations of integrating AI, AR, and robotics into Halal certification processes.

The study primarily focuses on Scopus-indexed articles, omitting other relevant sources such as IEEE, Web of Science, and Google Scholar, this might lead to potential bias.

The paper does not provide a conceptual framework integrating Industry 4.0 technologies into the Halal Meat Supply Chain (HMSC). It only mentioned the technologies. At least there should be a framework or suggestions.

What are the cost implications of using the Blockchain, AI, IoT, Digital Twins, AR?

What are the training requirements for workers? It should at least mention some guidelines and requirements

Based on your valuable comment, we have added section 6. as limitations

Thank you once again for your valuable feedback, which has helped us improve the quality of our work.